# A Counterfactual-Based Approach to Prevent Crowding in Intelligent Subway Systems

## Abstract

Today, the cities we live in are far from being truly smart: overcrowding, pollution, and poor transportation management are still in the headlines. With wide-scale deployment of advanced Artificial Intelligence (AI) solutions, however, it is possible to reverse this course and apply appropriate countermeasures to take a step forward on the road to sustainability. In this research, explainable AI techniques are applied to provide public transportation experts with suggestions on how to control crowding on subway platforms by leveraging interpretable, rule-based models enhanced with counterfactual explanations. The experimental scenario relies on agent-based simulations of the De Ferrari Hitachi subway station of Genoa, Italy. Numerical results for both prediction of crowding and counterfactual (i.e., countermeasures) properties are encouraging. Moreover, an assessment of the quality of the proposed explainable methodology was submitted to a team of experts in the field to validate the model.

## 1 Introduction

### 1.1 Background

According to the European Directorate-General for Mobility and Transport (Commission et al., 2022), the total passenger transport activities in the EU-27 was estimated in 4446 billion pkm (passenger kilometers) in 2022. This estimate includes intra-EU air and sea transport and of course ground transportation. Intra-EU air and intra-EU maritime transport contributed for 4% and 0.2%, respectively. Passenger cars accounted for 80.6%, powered two-wheelers for 2.3%, buses and coaches for 6.6%, railways for 5% and tram and subways only for 1.2%. It is therefore clear that European citizens still prefer to travel by their own means rather than use public transportation to avoid possible delays and overcrowding. Thus, it is extremely necessary to improve public services in order to make the population more inclined to use them with the aim of obtaining advantages both in terms of safety and environmental sustainability. As a matter of fact, Artificial Intelligence (AI) can play a central role in improving public transportation and managing the influx of passengers (Ushakov et al., 2022). This topic is growing in importance and the amount of available literature has increased rapidly in recent years. Below, we present and discuss the most relevant research, with specific focus on the use of AI for subway passenger flow monitoring.

### 1.2 Related works

By definition, a smart city is a system in which integrated technologies, AI and services are combined to improve the quality of life of the citizens. Within this extremely broad scope, smart transportation proposes advanced solutions for monitoring and improving mobility efficiency through the use of transportation means (Sundaresan et al., 2021). With improved methodologies, technological innovations, and an increase in the number of inhabitants in medium and large cities, scientific research on these issues has become of paramount importance. For example, in Tang et al. (2023) a synthetically augmented balanced dataset of smart-card data has been used to train a deep neural network for the prediction of hourly passenger boarding demand on buses. In Niyogisubizo et al. (2023), the authors propose a Wide Deep Learning model for crash severity prediction where SHapley Additive exPlanations (SHAP) technique is used to increase model transparency by assessing the importance of each input feature in predicting the model output. Focusing on subways, several solu-

tions for monitoring and controlling the flow of passengers have been recently proposed. In Zheng et al. (2023), a novel methodology to monitor the congestion around the platform screen doors of the Guangzhou Subway station based on real-time surveillance videos analysis and position entropy is proposed. The study reported in Yuan et al. (2023) developed a model to predict the pedestrian distribution in subway waiting areas based on the bacterial chemotaxis algorithm. Passenger flow estimation is one of the hot topics in this field, and as for example Feng et al. (2023) introduces a novel model that combines convolution layers and multi-head attention mechanism to provide better inter-trip services, integrating and forecasting the passenger flow of multi-level rail transit networks to improve the connectivity of different transport modes. Passenger flow is not only about optimizing transportation, but also about the impact it can make on the surrounding area, such as the design of new subway stations, as stated in Gong et al. (2019) or the monitoring of environmental sustainability, as studied in Park et al. (2022) and Wu et al. (2022). In addition, the recent COVID-19 pandemic emphasized the importance of controlling and monitoring passenger flow to prevent overcrowded situations in which it is impossible to maintain a safe distance (Lu et al., 2023). In this context, eXplainable AI (XAI) methods can be extremely helpful since they allow the decisions made by a certain black-box model to be interpreted, increasing trust in the use of prediction models (Ferrario & Loi, 2022). More importantly, XAI methods may allow quantitative characterization of crowding situations based on data-driven approaches. This can be extremely beneficial to public transport companies in order to take countermeasures based on the decisions provided by prediction models. Despite this, recent literature (e.g., Zhao et al. (2020); Zou et al. (2022)) mainly focuses on the use of XAI techniques to prioritize and select features based on their importance in passenger flow prediction, rather than providing quantitative suggestions potentially applicable in practice.

### 1.3  CONTRIBUTION

The main objective of this paper is to combine explainable-by-design and post-hoc XAI techniques for the short-term prediction of crowding conditions in specific subway areas (i.e., the platforms) using a dataset derived from simulations. To the best of our knowledge, this is the first work that combines rule-based interpretable models with counterfactual explanations to (i) predict possible crowding situations and (ii) suggest quantitative actions to prevent those situations based on what-if scenarios.

This preliminary analysis will focus on a simple but straightforward use case in the city of Genoa, Italy. The Genoa subway system is a double-track single line of 7.1 km (4.4 mi) that connects the two main valleys of Genoa (Val Bisagno to the northeast with the Brignole stop and Valpolcevera to the northwest with the Brin stop) via the city center. The analysis will be devoted to the prediction of potential crowding situations in the De Ferrari Hitachi subway station, located below the main square of the city. The application of the proposed methodology to a real problem highlights the contribution of the research, making possible future developments of fully reliable XAI countermeasures for crowd prevention in city subways.

All codes and tests are publicly available, but will be shared after double-blind review.

## 2  MATERIALS AND METHODS

### 2.1  DATASET

In this work, a dataset containing simulations of the De Ferrari Hitachi subway station of Genoa was used. The dataset contains 28 variables (summarized in the Appendix, Table 5) derived from 12696 simulations of 2 hours each. The simulations were generated using an agent-based model that allows to simulate the individual behavior of passengers and the interaction with other passengers and the surrounding environment, based on parameters measured on-site or agreed upon interactions with stakeholders. In particular, the range of input parameters was set based on field-assessed values on weekdays, during off-peak hours. This simulation approach proved very useful in generating a sufficiently large set of realistic simulated scenarios in a cheaper and less time consuming way with respect to on-field experimental data collection (Nikolenko, 2021). The dataset was used to characterise the parameters related to a situation of potential crowding and suggest which values to act on (quantitatively) in the short run, to obtain the alternative uncrowded scenario i.e., its counterfactual. In particular, we were interested in predicting the level of crowding on the two platforms of the

subway station (i.e., towards Brin and towards Brignole) during the last 15 minutes of simulation, that is, in the time window $[t, t + \Delta t]$, with $\Delta t = 15$ minutes. The input variables of the prediction model were extracted in the time window $[t - 2\Delta t, t]$, i.e., we considered the situation of the simulated subway station between 45 minutes and 15 minutes before the end of the simulation. Based on the simulated data, a *critical crowding threshold THR* of 30 people was selected and used as a discriminating value to identify the output of the classification problem. Having defined this threshold, 2 possible scenarios can thus be tested for each platform: average number of people waiting at the platform in the time window $[t, t + \Delta t]$ lower than *THR* (class 0) or greater than *THR* (class 1). Based on the available data, the following distributions of output classes result:

- platform towards Brin: 6509 simulations belonging to class 0, 6187 simulations belonging to class 1.
- platform towards Brignole: 11718 simulations belonging to class 0, 978 simulations belonging to class 1.

De Ferrari Hitachi subway station is only one stop away from Brignole station, therefore, a smaller number of critical cases (i.e., class 1 points) on the corresponding platform is considered plausible. A subset of 7 variables was selected to be used in the counterfactual analysis and denoted as $V1, \ldots, V7$. The subset of variables is listed in Table 5 of the Appendix. These variables were considered meaningful to ensure a trade-off between ability to represent the evolution of the crowding scenario and clarity of the explanation, based on preliminary interaction with transportation experts and feature ranking analysis.

## 2.2 EXPLAINABLE AI TECHNIQUES

### 2.2.1 LOGIC LEARNING MACHINE

The Logic Learning Machine (LLM) is an XAI method belonging to the family of transparent-by-design, global, rule-based models that provides a set of $n$ interpretable rules. The rule learning procedure can be summarized in four steps. First (Discretization), continuous inputs are discretized while maintaining a trade-off between number of discrete values and information retained. Then, the discretized values are converted into binary strings (Binarization) that are in turn used to extract a set of positive Boolean functions (Synthesis). Finally, the obtained Boolean functions are mapped into a set of rules (Rules extraction). Each rule is defined as an *if (premise) then (consequence)* statement, where premise is a logical AND of $m$ conditions $c_j$, and consequence is the assigned output class (Muselli & Ferrari, 2011). After computing $TP(R_i), FP(R_i), TN(R_i)$, and $FN(R_i)$ that are, respectively, the true positives, false positives, true negatives, and false negatives associated with a certain rule $R_i$, we can define two main measures of the goodness of that rule:

$$C(R_i) = \frac{TP(R_i)}{TP(R_i) + FN(R_i)} \tag{1}$$

$$E(R_i) = \frac{FP(R_i)}{FP(R_i) + TN(R_i)} \tag{2}$$

where $C(R_i)$ is the *covering* and $E(R_i)$ is the *error*. $C(R_i)$ measures the number of data samples that is correctly covered by $R_i$, whereas $E(R_i)$ measures the number of data samples that is wrongly covered by $R_i$ i.e., the number of samples that satisfies the premise of $R_i$ but belongs to a different output class. Thus, the greater the covering, the higher is the generality and the correctness of that rule and the larger is the error, the lower is the quality of the rule.

### 2.2.2 FEATURE AND VALUE RANKING

The importance of a feature in predicting the output can be derived from equation 1 and equation 2 by considering the rule conditions in which that feature is involved. Specifically, the importance of a condition (i.e., of the related feature) $Imp(c_j)$ can be calculated as:

$$Imp(c_j) = (E(R_i') - E(R_i))C(R_i) \tag{3}$$

by comparing the error of rule $R_i$, in which condition $c_j$ occurs, and the error of the same rule without that condition, that is $R_i'$. Features importance can then be ordered to provide a feature ranking. The same argument can be extended to intervals of values, thus giving rise to *value ranking*.

### 2.2.3 COUNTERFACTUAL EXPLANATIONS

Counterfactual explanations (from now on simply referred to as *counterfactuals*) belong to the family of local XAI techniques. In a binary classification problem, a counterfactual explanation is defined as the set of minimal changes that can be applied to the input features related to a specific record in order to change its predicted class. In other works (Lenatti et al., 2022; Carlevaro et al., 2022), the authors proposed an original methodology to construct counterfactuals from Support Vector Domain Description (SVDD) (Huang et al., 2011). Named $S_1$ and $S_2$ the two regions individuated by the SVDD, defined a subset of controllable features $\mathbf{u}$ and a subset of non-controllable features $\mathbf{z}$, so that a feature $\boldsymbol{x} \in \mathcal{X}$ can be thought as $\boldsymbol{x} = \left(u^1, u^2, \ldots, u^n, z^1, z^2, \ldots, z^m\right) \in \mathbb{R}^{n+m}$, the counterfactual model consists in finding the minimal variation ($\Delta\mathbf{u}$) of the controllable variables so that the feature $\boldsymbol{x} = (\mathbf{u}, \mathbf{z}) \in S_1$ moves to the opposite class $\boldsymbol{x}^* = (\mathbf{u} + \Delta\mathbf{u}^*, \mathbf{z}) \in S_2$. This implies the solution of the following optimization problem

$$\min_{\Delta\mathbf{u}\in\mathbb{R}^n} \quad d\big(\boldsymbol{x}, (\mathbf{u} + \Delta\mathbf{u}, \mathbf{z})\big)$$
$$\text{subject to} \quad \boldsymbol{x}^* \notin S_1 \quad \text{and} \quad \boldsymbol{x}^* \in S_2$$

where $d$ is a distance that best fits with the topology of the data. In other words, the *counterfactual* $\boldsymbol{x}^*$ is the nearest point, with respect to distance $d$, that belongs to the class opposite to the original class of a given point $\boldsymbol{x}$, taking into account that *only* controllable features $\mathbf{u}$ can be modified.

It is worth underling that the SVDD model for defining the input regions $S_1$ and $S_2$ was trained on all variables in the dataset to obtain a complete and accurate representation of the system. Then, counterfactuals were extracted only on $V1, \ldots, V7$ for better comprehensibility and visualization purposes, as pointed out in Section 2.1.

To make the counterfactual analysis more specific, three different, alternative counterfactual explanations were generated for each input observation, obtained by applying different constraint conditions to some of the input variables (i.e., imposing the no-variation condition to a subset of features, in the counterfactuals search algorithm):

- Unconstrained counterfactuals (C) are defined as the counterfactual explanations obtained without imposing any constraint on the input data, i.e., allowing all features to vary.

- Counterfactuals constrained on People-related features ($C_{CP}$) are defined as the counterfactual explanations obtained by constraining the features more strictly related to people flow, namely $V1$, $V2$, and $V7$.

- Counterfactuals constrained on Trains-related features ($C_{CT}$) are defined as the counterfactual explanations obtained by constraining the features related to trains, namely $V3$, $V4$, $V5$, and $V6$.

We remark that people-related features (i.e., $V1$, $V2$, and $V7$) are common to the 2 models, whereas train-related features (i.e., $V3, V4, V5$ and $V6$) depends specifically on each model, relate specifically to the platform to which the model refers. As an example, in Section 4 the counterfactual explanations of two different simulated scenarios are shown, one for each subway destination (Brin, Brignole). To quantitatively evaluate the proposed counterfactual explanations in terms of their ability to be distinguished from data points in the factual class discriminative power was calculated, as defined in Lenatti et al. (2022). The general structure of the methodology is summarized in the flowchart in Figure 1. During the training phase we collected simulation data in the time window $[t - 2\Delta t, t]$ and train a SVDD to learn the mapping function $f : \mathcal{X}_t \longrightarrow y_{t+\Delta t} \in \{0, 1\}$ between input and output. Then, during the operational phase, we use the information vector collected up to $\tilde{t}$ to forecast the crowding situation in the next time window, using the previously learnt mapping function $f$. If the prediction $f(\boldsymbol{x}_{\tilde{t}})$ is below $THR$ there is no need to control the system as the crowding situation is under control. Otherwise, if the prediction $f(\boldsymbol{x}_{\tilde{t}})$ is above $THR$, we can generate the counterfactual explanation of $\boldsymbol{x}_{\tilde{t}}$ and use the changes suggested by the latter to implement a data-driven control action to bring the system back toward a non-crowding situation. The actions of the counterfactual example will be visible in the subsequent time intervals, $\tilde{t} + \Delta\tilde{t}$ and $\tilde{t} + 2\Delta\tilde{t}$ depending on the changed variables.

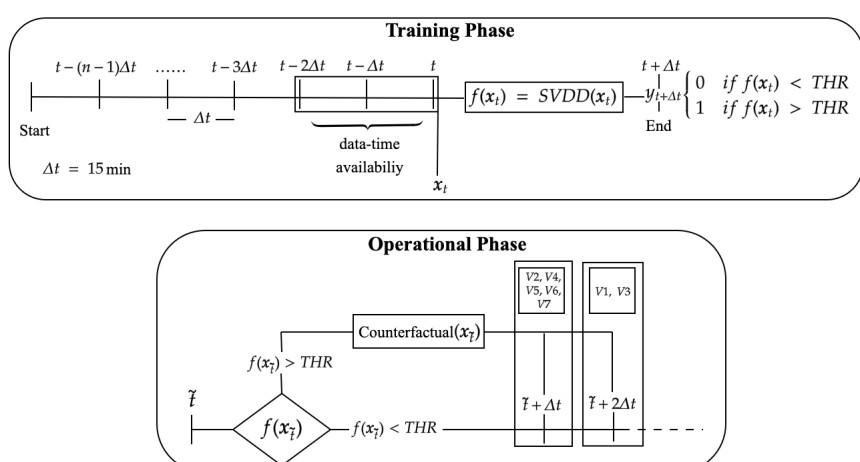

Figure 1: Methodology flow chart describing the SVDD training phase from a set of simulations (top panel) and the operational phase (bottom panel).

## 2.3 APPLICATION GROUNDED EVALUATION

XAI methods have shown great potential in increasing user confidence in automatic decision models, however, how to evaluate those techniques is still a matter of debate (Doshi-Velez & Kim, 2017). One of the most straightforward way is to perform an application grounded evaluation, that is, to assess the quality of explanations in their applicative context, involving domain experts (Jesus et al., 2021). A team of 5 experts in the field of transportation and logistics that possess only basic AI knowledge was asked to fill out a survey anonymously. The survey was delivered using Microsoft Forms. Respondents participated in the survey on a voluntary basis, with no incentive. The survey included different sections. First, the experts were asked to evaluate four scenarios showing the average values of variables $V1$-$V7$ for each specific output class and each specific model. The experts were blinded to the actual output class and were asked to select whether each scenario corresponded to a situation with a number of people on the platform below or above *THR*. They were also asked to specify their level of confidence on a 4-level scale. This first part of the questionnaire aimed to assess whether the chosen features and the output were considered sufficiently explanatory of the problem to be modeled. Then, the experts were asked to evaluate four examples of factuals with the corresponding counterfactuals C, $C_{CP}$ and $C_{CT}$ (2 related to Brin destination and 2 related to Brignole destination). For each example, the experts were asked to specify the level of agreement with the proposed suggestions on a scale of 1 to 5 and to provide an optional brief justification. In addition, the experts had to specify which of the 3 proposed solutions was considered the best. Finally, each expert was asked to assess the plausibiity and applicability of the results and to provide overall feedback on the proposed methodology. In addition, experts were asked to evaluate which features, among those considered in the model, are more easily controllable in the short run and to suggest any additional variables to be considered in a follow-up study.

## 3 RESULTS

### 3.1 LLM FOR CROWDING PREDICTION

Two separate LLMs (one per platform), were trained on 70% of the data and tested on the remaining 30%. Accordingly, we will refer to two distinct models: $LLM_{Bg}$ aims to predict the state of crowding on the platform in the Brignole direction, whereas $LLM_{Br}$ focuses on predicting crowding on the platform in the Brin direction. The classification performance via LLM (for each model) are reported in Table 1. Table 2 reports the main characteristics of $LLM_{Bg}$ and $LLM_{Br}$ in terms of number of decision rules, covering and error. $BgPP_{t-2\Delta t}$, $PIS_{t-2\Delta t}$, and $BgTI_{t-\Delta t}$ were particularly decisive in predicting the exceedance of *THR* in the $LLM_{Bg}$ model (i.e., feature ranking $> 0.2$), whereas the most relevant variables for the $LLM_{Br}$ model are $PIS_{t-2\Delta t}$, $PIE_{t-\Delta t}$, $PIS_{t-\Delta t}$, and $API_{t-\Delta t}$, all of which are variables closely related to the flow of passengers entering and circu-

Table 1: Performance results of $LLM_{Bg}$ and $LLM_{Br}$.

| Model | training accuracy | test accuracy | sensitivity (on test set) | specificity (on test set) |
|---|---|---|---|---|
| $LLM_{Bg}$ | 0.82 | 0.82 | 0.73 | 0.83 |
| $LLM_{Br}$ | 0.75 | 0.70 | 0.71 | 0.69 |

lating through the station in the 2 previous intervals. Feature ranking demonstrates once again how the number of passenger at the platform is largely influenced by the flow of passengers entering and stationing within the subway station in the 2 time intervals considered (i.e., $[t - 2\Delta t, t - \Delta t]$ and $[t - \Delta t, t]$), as well as by the trains frequency. The use of XAI techniques such as the LLM allows for a more in-depth exploration of these intuitive considerations, by providing quantitative cut-off values in the form of a value ranking. For example, the value ranking provides thresholds equal to 27 for $BgPP_{t-2\Delta t}$, 538 for $PIS_{t-2\Delta t}$, and 14 for $BgTI_{t-\Delta t}$ when applied to the $LLM_{Bg}$ model. This means that in general, values of these variables above the identified cut-off values are associated with a higher probability of providing an output of 1 in the model and therefore associated with a situation of potential crowding.

Table 2: Main characteristics of $LLM_{Bg}$ and $LLM_{Br}$: # of rules, covering and error.

| Model | # of rules | $C(\mathbf{R_i})$ (mean ±s.d.) | $E(\mathbf{R_i})$ (mean ±s.d.) |
|---|---|---|---|
| $LLM_{Bg}$ | 34 (23;11) | 11.00%± 8.14% | 4.6%± 0.51% |
| $LLM_{Br}$ | 50 (25;25) | 7.20%± 4.46% | 4.77%± 0.65% |

Table 3: Mean and standard deviation of the changes suggested by C, $C_{CP}$ and $C_{CT}$ for variables $V1, \ldots, V7$.

| Feature | Brin | | | Brignole | | |
|---|---|---|---|---|---|---|
| | C | $C_{CP}$ | $C_{CT}$ | C | $C_{CP}$ | $C_{CT}$ |
| $V1$ | -174 ±127 | - | -287 ±167 | -115 ±123 | - | -254 ±185 |
| $V2$ | -84 ±129 | - | -131 ±200 | -23 ±116 | - | -41 ±256 |
| $V3$ [s] | -16 ±155 | -8 ±270 | - | -48 ±158 | 0 ±271 | - |
| $V4$ [s] | -197 ±170 | -281±237 | - | -178 ±157 | -247 ±255 | - |
| $V5$ | -16 ±77 | -30±119 | - | 3 ±78 | -9 ±118 | - |
| $V6$ | 9 ±34 | 17±47 | - | 6 ±17 | 5 ±24 | - |
| $V7$ | 0 ±14 | - | 0 ±22 | 4 ±14 | - | 0 ±24 |

## 3.2 EVALUATION OF COUNTERFACTUAL EXPLANATIONS

### 3.2.1 QUANTITATIVE EVALUATION

A set of factuals was extracted from test records having output equal to 1 (i.e., 1051 for the Brin travel direction and 214 for the Brignole travel direction) and counterfactual explanations for each of the three typologies described in Section 2.2.3 were generated for each factual. The discriminative power of counterfactual explanations generated for the Brin travel direction was of about 90.6%, 91.8%, and 93.9% for C, $C_{CP}$ and $C_{CT}$, respectively. The discriminative power of counterfactual explanations generated for the Brignole travel direction was on average slightly lower compared

to that of Brin (86.7%, 94.5%, and 89.8% for C, $C_{CP}$ and $C_{CT}$, respectively). Table 3 reports the average changes in $V1, \ldots, V7$ as suggested by C, $C_{CP}$ and $C_{CT}$, together with the corresponding standard deviation. For the sake of clarity, variables $V3$ and $V4$ have been reported in seconds instead of minutes as in the rest of the article. These values suggest what is the global trend that each variable needs to observe in order to move toward a non-crowded situation. However, these average values are only indicative, as they could differ significantly depending on the specific observation considered, i.e., depending on the specific values of the factual. The counterfactual explanations require on average to reduce $V1, \ldots, V5$, to slightly increase $V6$, and to not intervene on $V7$. In general, when some variables are constrained like in $C_{CP}$ and $C_{CT}$, the remaining controllable variables vary more significantly, as it can be seen by an increase in the absolute value of the mean change and a greater standard deviation.

Table 4: Average values of variables $V1, \ldots, V7$ on the training set, for each specific output class and each specific model.

|  | Brin | | Brignole | |
| --- | --- | --- | --- | --- |
| **Feature** | **A** | **B** | **C** | **D** |
| $V1$ | 418 | 280 | 342 | 477 |
| $V2$ | 386 | 316 | 345 | 386 |
| $V3$ | 10 | 10 | 10 | 10 |
| $V4$ | 11 | 9 | 10 | 14 |
| $V5$ | 218 | 201 | 173 | 174 |
| $V6$ | 42 | 32 | 11 | 22 |
| $V7$ | 38 | 38 | 38 | 38 |
| Output | 1 | 0 | 0 | 1 |

## 3.3 APPLICATION GROUNDED EVALUATION

The average survey completion time was 18 minutes. Despite reporting minimal or basic knowledge in AI, respondents believe that AI will play a pivotal role in crowd management in public environments. In the first series of questions the experts were asked to select the crowding class (i.e., 0 or 1) given a set of 7 features ($V1$–$V7$) describing a specific scenario, as shown in Table 4. In general, crowded scenarios (output=1, case A and D) show higher $V1, V2, V4, V5, V6$ with respect to non-crowded scenarios (output=0, case B and C), whereas $V3$ and $V7$ are similar in both scenarios, for both platforms. All experts correctly assessed case A as a situation where the number of people on the platform is above the threshold, stating fairly high (3 out of 5) or high (2 out of 5) confidence in the answer given. Similarly, 4 out of 5 experts correctly assessed case D, although with a decrease in reported confidence (low confidence: 2; fairly high confidence: 2; high confidence: 1). As for non-crowded scenarios, Case B was correctly classified by 3 out of 5 experts (low confidence: 2; fairly high confidence: 3), whereas case C was correctly classified only by 2 out of 5 experts (low confidence: 1; fairly high confidence: 4). In 3 out of 4 examples, the experts correctly predicted the output class; the output class 1 was predicted more accurately, by an higher number of experts, although they were rarely completely confident in the answer given.

Then, the experts were asked to evaluate a set of counterfactual explanations. One example related to the platform in Brin direction is reported in Figure 2. Referring to this example, the majority of experts were found to agree with the proposed suggestions (moderately agree: 3; neither agree nor disagree: 1; moderately disagree: 1). $C_{CP}$ was judged to be the most realistic solution, as it suggests preventing a crowded situation on the platform by reducing $V3$ and $V4$ by 3 minutes, that is, reducing the interval between trains in the previous two time windows. Furthermore, the presence of fewer people on the platform at time $t-1$ (lower $V6$) is associated with a lower probability of crowding at time $t$. In contrast, counterfactual C was considered counter intuitive by one of the experts, since the passengers inflow in the previous two time intervals ($V1$ and $V2$) is reduced, but

at the same time there is an increase in the number of people waiting at the platform ($V6$). This might suggest that the crowding condition is related to the combination of passengers on the stairs and at the platform rather than the number of passengers in a specific station area.

In general, the proposed counterfactual explanations were considered realistic by all the experts, however they were not always considered readily applicable (realistic and applicable: 3; realistic but not applicable: 2). Among the variables considered in the simplified simulation scenario, the passengers inflow was rated as the most controllable variable in the short-run (15-30 minutes) (4 votes), followed by the number of people boarding the train and train frequency (2 votes each). According to the surveyed experts, the countermeasures deemed most effective in achieving the values suggested by the counterfactual explanations include turnstiles blockage to reduce station entrances and a reorganization of the timetable to adjust time intervals between consecutive trains. Additional suggested controllable variables include the waiting time at the platform, the number of carriages per train and the train length of stay at the station.

## 4 DISCUSSION

### 4.1 LLM FOR CROWDING PREDICTION

The use of historical data for short-term passenger flow prediction has proved of paramount importance for efficiently improving subway system management (Wei et al., 2023). In this work, LLM has shown the ability to predict the evolution of crowding in a given station area (i.e., a specific subway platform, in this case) by having information on the incoming, outgoing, and current passenger flow of the platforms in a previous time window. Prediction accuracy can be considered satisfactory, with values above 80% when considering $LLM_{Bg}$ and slightly lower values (around 70%) when considering $LLM_{Br}$. The two models are characterized by a quite high number of rules that can sufficiently represent both classes, with a covering that can reach up to 30% and an error associated with individual rules lower than 5%. Rule-based models can be further refined by filtering out redundant rules or conditions and merging similar rules, allowing the logic underlying knowledge extraction to be streamlined while maintaining satisfactory predictive performance.

Rule-based approaches have been already used for passenger flow prediction. For example, Zhao et al. (2020) explored the influence of temporal, spatial and external features in predicting daily passenger flow using tree based ensemble methods (random forest and gradient boosting decision tree) on data from the Shanghai Metro Automatic Fare Collection system. However, feature ranking was used only for feature selection purposes. In our work, the analysis of feature ranking allowed to identify the main features that the model uses to predict a particular output, whereas the further value ranking analysis allowed to quantitatively specify the values of those features that are most determinant for a certain output. In particular, the value ranking given for $LLM_{Bg}$ in Section 3.1 are similar or slightly higher with respect to the average values for output equal to 1 (e.g., V2 and V4 in

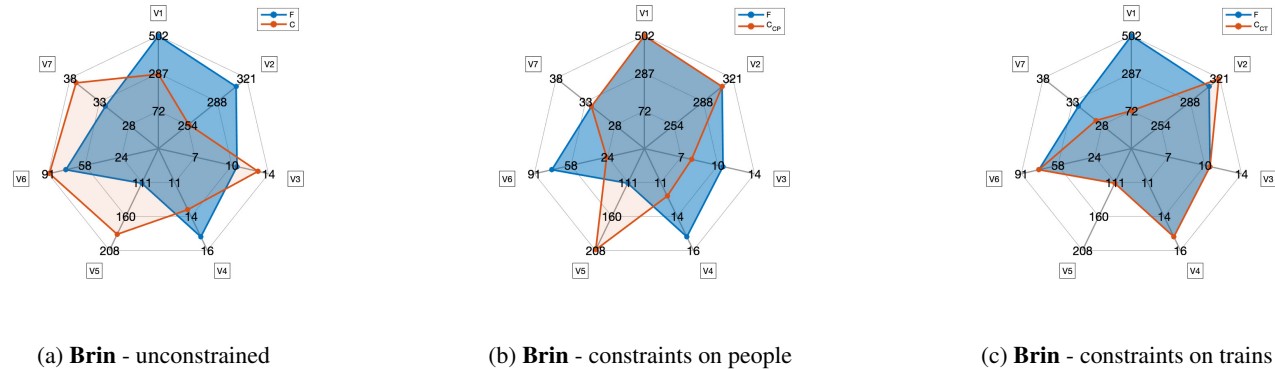

(a) **Brin** - unconstrained      (b) **Brin** - constraints on people      (c) **Brin** - constraints on trains

Figure 2: Spiderplot of the three proposed scenarios containing the factuals and respective counterfactuals for the platform in Brin direction.

case D, Table 4) but definitely higher with respect to the average values for output equal to 0 (e.g., V2 and V4 in case C, Table 4), thus showing high discrimination capabilities between the two classes. Although this analysis has enabled the identification of global discrimination cut-off values related to individual features, the user could benefit from additional analyses of individual scenarios through local explanations. Thus, an extremely useful tool is the generation of counterfactual explanations that provide quantitative suggestions by varying multiple features simultaneously while focusing on a single scenario.

## 4.2 Counterfactual explanations for crowding prevention

The quality of the set of counterfactual explanations was verified both quantitatively through the calculation of discriminative power and qualitatively by consulting expert opinion by means of a questionnaire. The discriminative power is around 90% for C, $C_{CP}$ and $C_{CT}$ in both platforms, hence, the set of explanations belonging to class 0 can be accurately distinguished from the source class of factuals (class 1). Moreover, we can observe that the average changes suggested by the counterfactuals (Table 3) are consistent with the distributions of the training input data in the 2 classes (Table 4). However, to verify the actual applicability of the method this metric was not sufficient and interaction with experts was necessary. According to the experts, the suggestions produced through counterfactual explanations can be considered as realistic, however, in the future it might be useful to consider additional controllable features, such as train dwell time at the station and the number of carriages per train which could possibly be added if the station is expected to be significantly crowded. An additional interesting insight that emerged from the questionnaire is that the suggested changes may not systematically be applicable in the short run, as the logistic infrastructure may not be able to intervene quickly enough (e.g., increase train capacity, dynamically control station access). This aspect was in part considered through the introduction of different explanations focusing on different subgroups of features and can be further developed through iterative interaction with the stakeholders.

## 4.3 Limitations and future research

In this study, the method was applied to a specific station location, but it can be easily generalized to other areas of the station such as entrances, and emergency exits. Moreover, in this preliminary study, a relatively low critical crowding threshold (30 people on the platform) was chosen based on considerations due to the chosen facility and its normal passenger flow. In fact, the objective of the study is to predict potential crowding in everyday situations, in the short term, whereas the presence of exceptional events with excessively higher than normal flows (e.g., events, concerts, soccer games) is known with due advantage and managed differently. However, it is important to note that the proposed analysis may be easily applied to different threshold values. Future developments of the study may cover different aspects, such as the extension of the prediction window to consider possible inner dynamics in the medium to long term, the comparison of counterfactual explanations obtained with different critical crowding threshold levels or the customization of the set of controllable and non-controllable features defined based on requirements established together with the transportation infrastructure stakeholders. Furthermore, expert comments highlighted the need to analyze the causal relationships between variables in order to obtain more realistic suggestions.

## 5 Conclusion

Encouraging the use of public transportation by improving infrastructures and passenger flow management is one of the main steps to promote environmental sustainability. From this perspective, our research focused on the analysis of passenger flow at subway stations through explainable AI, particularly rule-based models and counterfactual explanations. A specific use case in the city of Genoa was selected for this purpose. Besides quantitative evaluation, the proposed explanations were preliminarily assessed by a team of experts in the field of transportation, in terms of their realism and applicability. Results suggest that counterfactual explanations may provide interpretable insights that can be used as a reference point for experts in the decision-making process when developing countermeasures for efficient crowd management.

## 6 REPRODUCIBILITY STATEMENT

All examples and results shown in this article are fully reproducible, but the code will be shared after the double-blind review phase.
Below is a detailed description of the dataset used for model evaluation.

Table 5: Dataset features capturing the two time intervals of interest: minimum value, maximum value, mean, and standard deviation. $V1, \ldots, V7$, denotes the variables used in the counterfactual analisys.

| | Name | Min | Max | Mean | Std | Description |
|---|---|---|---|---|---|---|
| **Common** | $\text{PIS}_{t-2\Delta t}$ | 36 | 711 | 347 | 176 | Passenger **I**nflow from **S**tairs in the time window $[t-2\Delta t, t-\Delta t]$, [passengers/h] (**V1**) |
| | $\text{PIS}_{t-\Delta t}$ | 37 | 713 | 345 | 178 | Passenger **I**nflow from **S**tairs in the time window $[t-\Delta t, t]$, [passengers/h] (**V2**) |
| | $\text{PIE}_{t-2\Delta t}$ | 0 | 7 | 4 | 2 | Passenger **I**nflow from **E**levator in the time window $[t-2\Delta t, t-\Delta t]$, [passengers/h] |
| | $\text{PIE}_{t-\Delta t}$ | 0 | 8 | 3 | 2 | Passenger **I**nflow from **E**levator 15 minutes before, [passengers/h] |
| | $\text{API}_{t-2\Delta t}$ | 3 | 79 | 38 | 14 | **A**verage number of **P**assengers on the **I**ntermediate level in the time window $[t-2\Delta t, t-\Delta t]$, [passenger] |
| | $\text{API}_{t-\Delta t}$ | 4 | 78 | 38 | 14 | **A**verage number of **P**assengers on the **I**ntermediate level in the time window $[t-\Delta t, t]$, [passenger] |
| | $\text{MPI}_{t-2\Delta t}$ | 1 | 24 | 8 | 3 | **M**aximum number of **P**assengers on the **I**ntermediate level in the time window $[t-2\Delta t, t-\Delta t]$ [passenger] |
| | $\text{MPI}_{t-\Delta t}$ | 1 | 24 | 8 | 3 | **M**aximum number of **P**assengers on the **I**ntermediate level in the time window $[t-\Delta t, t]$, [passenger] (**V7**) |
| **Brignole** | $\text{BgTI}_{t-2\Delta t}$ | 5 | 15 | 10 | 4 | Average **T**ime **I**nterval between trains on the platform in Brignole direction in the time window $[t-2\Delta t, t-\Delta t]$, [min] (**V3**) |
| | $\text{BgTI}_{t-\Delta t}$ | 5 | 15 | 10 | 4 | Average **I**nterval between **T**rains on the platform in Brignole direction in the time window $[t-\Delta t, t]$, [min](**V4**) |
| | $\text{BgPB}_{t-2\Delta t}$ | 8 | 412 | 190 | 81 | Average number of **P**assengers on the train **B**efore the stop on the platform in Brignole direction in the time window $[t-2\Delta t, t-\Delta t]$, [passenger] |
| | $\text{BgPB}_{t-\Delta t}$ | 6 | 412 | 193 | 80 | Average number of **P**assengers on the train **B**efore the stop on the platform in Brignole direction in the time window $[t-\Delta t, t]$, [passenger] (**V5**) |
| | $\text{BgPGO}_{t-2\Delta t}$ | 7 | 71 | 37 | 16 | Average number of **P**assengers **G**etting **O**ff the train on the platform in Brignole direction in the time window $[t-2\Delta t, t-\Delta t]$, [passenger] |
| | $\text{BgPGO}_{t-\Delta t}$ | 7 | 72 | 36 | 17 | Average number of **P**assengers **G**etting **O**ff the train on the platform in Brignole direction in the time window $[t-\Delta t, t]$, [passenger] |
| | $\text{BgTA}_{t-2\Delta t}$ | 1 | 414 | 193 | 82 | Average number of passengers on the **T**rain **A**fter departing from De Ferrari station in Brignole direction in the time window $[t-2\Delta t, t-\Delta t]$, [passenger] |
| | $\text{BgTA}_{t-\Delta t}$ | 1 | 414 | 170 | 81 | Average number of passengers on the **T**rain **A**fter departing from De Ferrari station in Brignole direction in the time window $[t-\Delta t, t]$, [passenger] |
| | $\text{BgPP}_{t-2\Delta t}$ | 0 | 95 | 12 | 11 | Average number of **P**assengers waiting at the **P**latform in Brignole direction in the time window $[t-2\Delta t, t-\Delta t]$, [passenger] |
| | $\text{BgPP}_{t-\Delta t}$ | 0 | 109 | 13 | 12 | Average number of **P**assengers waiting at the **P**latform in Brignole direction in the time window $[t-\Delta t, t]$, [passenger] (**V6**) |
| **Brin** | $\text{BrTI}_{t-2\Delta t}$ | 5 | 15 | 10 | 4 | Average **T**ime **I**nterval between trains on the platform in Brin direction in the time window $[t-2\Delta t, t-\Delta t]$, [min] (**V3**) |
| | $\text{BrTI}_{t-\Delta t}$ | 5 | 15 | 10 | 4 | Average **T**ime **I**nterval between trains on the platform in Brin direction in the time window $[t-\Delta t, t]$, [min] (**V4**) |
| | $\text{BrPB}_{t-2\Delta t}$ | 1 | 412 | 180 | 87 | Average number of **P**assengers on the train **B**efore the stop on the platform in Brin direction in the time window $[t-2\Delta t, t-\Delta t]$, [passenger] |
| | $\text{BrPB}_{t-\Delta t}$ | 1 | 412 | 180 | 87 | Average number of **P**assengers on the train **B**efore the stop on the platform in Brin direction in the time window $[t-\Delta t, t]$, [passenger] (**V5**) |
| | $\text{BrPGO}_{t-2\Delta t}$ | 0 | 16 | 8 | 17 | Average number of **P**assengers **G**etting **O**ff the train on the platform in Brin direction in the time window $[t-2\Delta t, t-\Delta t]$, [passenger] |
| | $\text{BrPGO}_{t-\Delta t}$ | 0 | 17 | 7 | 17 | Average number of **P**assengers **G**etting **O**ff the train on the platform in Brin direction in the time window $[t-\Delta t, t]$, [passenger] |
| | $\text{BrTA}_{t-2\Delta t}$ | 3 | 415 | 209 | 87 | Average number of passengers on the **T**rain **A**fter departing from De Ferrari station in Brin direction in the time window $[t-2\Delta t, t-\Delta t]$, [passenger] |
| | $\text{BrTA}_{t-\Delta t}$ | 3 | 414 | 211 | 88 | Average number of passengers on the **T**rain **A**fter departing from De Ferrari station in Brin direction in the time window $[t-\Delta t, t]$, [passenger] |
| | $\text{BrPP}_{t-2\Delta t}$ | 0 | 213 | 36 | 26 | Average number of **P**assengers waiting at the **P**latform in Brin direction in the time window $[t-2\Delta t, t-\Delta t]$, [passenger] |
| | $\text{BrPP}_{t-\Delta t}$ | 0 | 216 | 37 | 26 | Average number of **P**assengers waiting at the **P**latform in Brin direction in the time window $[t-\Delta t, t]$, [passenger] (**V6**) |

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
