# OpenReview forum: "A counterfactual-based approach to prevent crowding in intelligent subway systems"
_ICLR.cc/2024/Conference — Submitted to ICLR 2024_

### Official Review · Reviewer_fGpu · 2023-10-21

**Soundness:** 2 fair
**Presentation:** 3 good
**Contribution:** 2 fair
**Rating:** 3
**Confidence:** 3

**Summary:**

This paper discusses a counterfactual based approach to optimal control of the parameters of a system, such that the observed situation can be changed. They discuss a case study of crowding in a train station, for which data is generated using a large number of agent-based simulations under different control settings. The authors use Logic Learning Machine (LLM) to evaluate different interpretable rules based on an SVDD classifier trained on the simulated data. Then they rank the features according to this LLM analysis and finally identify the optimal set of features to change, to achieve the objective (i.e. convert a situation from "high crowding" to "low crowding")

**Strengths:**

An interesting problem with a good use-case.

**Weaknesses:**

1) lack of technical contributions in AI
2) too specific to a particular application
3) it is not clear if the proposed approach can be generalized to solve a larger class of problems related to policy framing

Regarding the optimization problem for calculating the counterfactuals, it is not clear how it is solved. It is not even clear if it has been formulated correctly.

**Questions:**

1) Regarding the optimization problem for counterfactuals, shouldn't the first constraint be x \in S1, rather than x* \notin S2?
2) How was it solved?

---

### Official Review · Reviewer_By14 · 2023-10-24

**Soundness:** 2 fair
**Presentation:** 1 poor
**Contribution:** 1 poor
**Rating:** 3
**Confidence:** 3

**Summary:**

The paper uses AI, specifically Logic Learning Machines (LLMs) - not to be confused with Large Language Models (also LLMs) - and counterfactual reasoning to try to predict congestion conditions in a railway station in the city of Genoa. To assess the predictive power of the models and to produce explainable results, the paper uses 7 variables and also compares the derived predictions with the opinions of 5 experts. By changing the values of these variables that are on of general, passenger-related or train-related, the study produces counterfactual scenarios that change from congested to uncongested conditions and vice versa. Results suggest general agreement between predictions and expert opinions.

**Strengths:**

- The paper tries to address an important problem, that of using AI to improve public transportation.
- The study is aware of the importance of explainability when using AI methods and tries to build an explainable AI method.

**Weaknesses:**

- The paper was very hard for me to understand. Many things are missing, e.g., the structure of the models that were used in the study, or are impossible to verify, e.g., the qualifications of the experts, the parameters of the simulations. Indicatively, the main variables are described in the 1-page appendix.
- The results seem preliminary and there are no signs of generalization beyond the current setting.
- From the current exposition, the exact technical contribution of the paper, if any, was unclear to me. It seems that the paper uses existing techniques (e.g., pre-trained LLM models), in a specific application. A large part of the paper is devoted to explaining things from the literature that are already known.

**Questions:**

I think that this paper is not ready for publication at its current state. My suggestion to the authors is to significantly improve the presentation and provide rigorous justifications for the background and steps used in it. E.g., the questionnaires are not available, the architecture of the code is not clear (even if the code cannot be disclosed due to anonymity) etc. Further, I suggest the focus to be on the novel methodological and conceptual contributions of the paper in a way that is transparent to the readers rather than on spending so much time on known things (e.g., precision and recall). Finally, the background and choices need to be specified accurately. Currently, we don't have much intuition about the simulation environment and the choices made for the parameters. E.g., in page 3 we read "Based on the available data,..." but it is unclear where this data refers to. If it is the simulation data, then, it is not justified why this distribution was chosen. The tables are hard to parse, with captions that don't explain the results, and no highlights (e.g., bold letters) in important findings.

Finally, it seems to me that since there are no methodological contributions, even if the above set of improvements is implemented, the paper is not a good fit for ICLR.

---

### Official Review · Reviewer_uUGu · 2023-10-31

**Soundness:** 1 poor
**Presentation:** 1 poor
**Contribution:** 1 poor
**Rating:** 1
**Confidence:** 3

**Summary:**

The paper studies applications of XAI methodology including Logic Learning Machine and Counterfactual Explanations on a simulated experimental subway station scenario.

**Strengths:**

The authors completed an application grounded evaluation that involved human experts' evaluating the AI performance.

**Weaknesses:**

My major concerns are with the contributions and the presentation of the paper. It reads like a course report rather than a quality research paper.

1. The claimed contribution in combining the two methods seem invalid to me. Essentially, there are two separate tasks (feature selection and counterfactual) and the two methods are used for them separately. To obtain counterfactual explanations, a different ML model (SVDD) is trained to predict the crowding. The two tasks and the two methods are only loosely connected under the same big problem.

2. The authors do not provide sufficient justification why the specific methods are needed and suitable for the underlying problem.
- Why do you adopt this particular rule based model not other models? You mentioned in Section 4.1 that others used tree-based ensemble methods. Why don't you use these methods?  It seems that LASSO or other methods are also explainable and can serve the feature selection tasks.
- Why do you need a transparent model? Evaluation by experts mostly justified the correctness of the feature selection but did not jusitify the need for a transparent explainable model. It looks that a deep black-box model can replace the rule based model without affecting any  further discussion.
- The counterfactual C was counter-intuitive as rated by the experts which weaken the interpretability of the model, yet the authors does not provide convincing reasons how to act on this problem. It is also unclear from the evaulation from experts whether the XAI stands out compared to other non-interpretable AI methods.

3. The details of simulation are missing, and the claimed significance through simulated data is doubted. It seems that there are a lot of specifications beind the scenes. How do you generate the variables? Why do you run 12696 simulations? Do you simulate the trains as agents or other type of objects? Why is your simulation good enough to replicate the scenarios in the real station? Why the other 21 variables are not meaningful to the counterfactual analysis?

4. The paper is poorly written.
- The authors spend pages to discuss well-established methods without any claimed novelty.
- The tables are meaningless: for example it is unclear what the authors want to convey through Table 4; the values of variables are not interesting to the audience because they are specific to the simulated problem.
- The interpretation of results are weak. For example, the covering looked not high and error not low, as displayed in Table 2. Do these numbers jusitfy the claimed performance? Similarly, are the accuracies in Table 1 the best we can achieve?


Lastly, I agree on most of the limitations that the authors identified themselves and I believe these limitations should be addressed before the paper is complete for submission.

**Questions:**

1. Is critical crowding threshold THR a justified notion? Has such binary model been used in study the crowdedness problem in existing literature? Or it oversimplified the problem.

2. Is the dataset too simplified to demonstrate the power of XAI? Can your methodology be easily extended to similar problems with more dimensions of complexity?

3.  The title mentioned "prevent crowding" yet no real action is provided. How does the counterfactual translates to the real decisions that can prevent the crowdedness? Did you do subsequent simulation that confirm the preventing power?

---

### Official Review · Reviewer_UPeU · 2023-11-06

**Soundness:** 3 good
**Presentation:** 3 good
**Contribution:** 2 fair
**Rating:** 6
**Confidence:** 1

**Summary:**

The paper proposes an explainable AI approach to prevent crowding in subway platforms by combining rule-based models and counterfactual explanations. The authors apply their method to a simulated scenario of the De Ferrari Hitachi subway station in Genoa, Italy, and evaluate its performance and quality with experts in the field of transportation. The paper aims to contribute to the literature on passenger flow prediction and management in public transportation systems.

**Strengths:**

- The paper addresses a relevant and timely problem of improving public transportation and sustainability by leveraging advanced AI solutions.
- The paper adopts a counterfactual-based approach that can provide actionable suggestions on how to change controllable features to achieve a desired outcome (i.e., reducing crowding).
- The paper uses realistic simulation data derived from an agent-based model that captures the individual behavior and interaction of passengers in the subway station.
- The paper evaluates the quality of the proposed explanations both quantitatively, by computing discriminative power, and qualitatively, by conducting an application-grounded evaluation with domain.
- The paper discusses the limitations and future research directions of their approach, such as extending the prediction window, comparing different threshold levels, customizing controllable features, and analyzing causal relationships.
- The authors state that they will open source the code after the double-blind review.

**Weaknesses:**

- The paper could provide more details on how the agent-based model was developed and validated, as well as how the input parameters were set based on field measurements or expert opinions.
- The paper could compare their approach with other existing methods for passenger flow prediction or crowd management in subways, such as deep learning models or optimization algorithms.
- The paper could report more examples of counterfactual explanations for different scenarios and platforms, and analyze their consistency and robustness.

**Questions:**

- How did you select the critical crowding threshold of 30 people? How sensitive is your method to different threshold values?
- How did you measure the level of confidence and agreement of the experts in your survey? What were the main sources of uncertainty or disagreement among them?
- How would you evaluate the impact or effectiveness of implementing your counterfactual suggestions in practice? What metrics or indicators would you use?

---

### Meta-Review · Area_Chair_kHgS · 2023-12-06

**Metareview:**

Reviewers were concerned that this paper may be too incremental since it combines a couple well-known techniques to solve a rather niche problem.

**Justification For Why Not Higher Score:**

No one really argued that the paper should be accepted. The only reviewer to give a positive score also wrote a note to the area chair specifically to say that they did not feel they had the expertise to review it.

**Justification For Why Not Lower Score:**

N/A

---

### Decision · Program_Chairs · 2024-01-16

Reject